# Source Localisation Using Wavefield Correlation-Enhanced Particle Swarm Optimisation

**George Rossides** [1,2], **Alan Hunter** [3] **and Benjamin Metcalfe** [2,*]

1 Marine Robotics Innovation Centre, Cyprus Marine and Maritime Institute, Larnaca 6023, Cyprus; george.rossides@cmmi.blue or g.rossides@bath.ac.uk
2 Department of Electronic & Electrical Engineering, University of Bath, Bath BA2 7AY, UK
3 Department of Mechanical Engineering, University of Bath, Bath BA2 7AY, UK; a.j.hunter@bath.ac.uk
* Correspondence: b.w.metcalfe@bath.ac.uk

**Abstract:** Particle swarm optimisation (PSO) is a swarm intelligence algorithm used for controlling robotic swarms in applications such as source localisation. However, conventional PSO algorithms consider only the intensity of the received signal. Wavefield signals, such as propagating underwater acoustic waves, permit the measurement of higher order statistics that can be used to provide additional information about the location of the source and thus improve overall swarm performance. Wavefield correlation techniques that make use of such information are already used in multi-element hydrophone array systems for the localisation of underwater marine sources. Additionally, the simplest model of a multi-element array (a two-element array) is characterised by operational simplicity and low-cost, which matches the ethos of robotic swarms. Thus, in this paper, three novel approaches are introduced that enable PSO to consider the higher order statistics available in wavefield measurements. In simulations, they are shown to outperform the standard intensity-based PSO in terms of robustness to low signal-to-noise ratio (SNR) and convergence speed. The best performing approach, cross-correlation bearing PSO (XB-PSO), is capable of converging to the source from as low as −5 dB initial SNR. The original PSO algorithm only manages to converge at 10 dB and at this SNR, XB-PSO converges 4 times faster.

**Keywords:** particle swarm optimisation; source localisation; marine swarm robotics; wavefield correlation

## 1. Introduction

Swarm robotics is the discipline of cooperative robotics, concerned with developing emergent collaborative behaviour in large numbers of robots. Robotic swarms are characterised by flexibility, scalability and robustness [1], as well as simplicity and low-cost, making them well suited for use in large, unexplored environments where semi-disposable systems are needed. For this reason, they have been applied to marine robotic applications including environmental monitoring [2], source detection of underwater oil spills [3] and exploration [4].

Robotic swarms are typically controlled using *swarm intelligence algorithms* (SIAs) [5]. One of the most common SIAs is particle swarm optimisation (PSO), which was inspired by the collective motion of birds and fish [6] and operates by assigning a fitness or cost to different locations. The robots are then drawn towards those locations. There have been numerous studies on the characteristics of PSO, leading to variants that address different limitations, depending on the application. Other SIAs include glowworm swarm optimisation (GSO) [7], which assigns a fitness value on each robot of the swarm, instead of locations in space; ant colony optimisation (ACO) [8], which mimics the foraging strategies employed by ants through the use of pheromones; and the firefly algorithm (FA), [9] which is similar to GSO, but each robot is attracted to all other robots and the strength of the attraction depends on the distance between robots.

PSO converts local signal intensity collected by sensors on each robot into location fitness (or cost). The robots communicate this information among the swarm and each then changes its motion based on both the local and collective knowledge of intensity, in order to converge to the source. PSO has been used in this way in applications such as olfaction source localisation [10–12]. Additionally, a variety of PSO variants have been proposed that enable PSO to control physical robotic swarms. Such techniques are not concerned with the way that the location of the source is identified and instead focus on controlling the motion of the robots of the swarm, by accommodating for underactuated motion, other physical motion constraints like maximum velocity and acceleration, communication constraints, and collision and obstacle avoidance. Robotic PSO (RPSO) is one such variant that incorporates obstacle avoidance into standard PSO [13]. Extended versions of RPSO also consider additional physical constraints of the robots [14,15]. Distributed PSO (dPSO) is another PSO variant that is instead concerned with minimising the information exchanged between the robots of the swarm [16]. Extensions of this algorithm therefore primarily focus on maintaining communication between the robots and possible ground stations or other communication points [17,18]. Finally, Micro-PSO ($\mu$PSO) is a variant that was designed to operate with a smaller number of robots and which has been adapted for use in simultaneous localisation and mapping (SLAM) tasks [19]. All of the aforementioned PSO variants have been implemented using intensity-based fitness assignment. Nevertheless, alternative ways of fitness assignment can be also employed to enhance source localisation performance.

Acoustic sensing is the primary method for source localisation in marine applications due to the efficient propagation of underwater sound waves. Sounds emitted by targets (such as marine mammals or submersible vehicles) propagate through the water and may be sensed by transducers such as hydrophones (e.g., passive SONAR). Akin to olfactory sensing, the intensity of acoustic fields can be used directly in PSO to control robotic swarms for the purpose of source localisation. However, acoustic fields contain rich spatio-temporal information (and thus higher-order statistics) that can be extracted using wavefield correlation techniques [20,21]. The measurement of field intensity as a function of direction can be performed using a multi-sensor array; for example the cross-correlation between two sensors provides directional information that can assist in the localisation of acoustic sources [22]. Two sensors are the fewest needed to perform cross-correlation, and whilst more sensors can improve the direction finding performance this is counter to the ethos of swarm robotics.

Thus, this paper introduces three novel PSO fitness-assignment approaches that use the correlation of signal readings from two sensors fitted to each robot. The first approach assigns fitness to a location based on only the correlation of the readings from each sensor, whilst the second and third modifications estimate the direction towards the source, thus exploiting the properties of acoustic wavefields to improve overall performance.

Section 2 of this paper demonstrates the original PSO algorithm as applied to source localisation, wherein the source emits a continuous signal of constant average intensity. In Section 3, three new PSO fitness-assignment approaches are introduced, detailing the theory of operation and the expected advantages. Section 4 introduces a simulation environment that is used to assess the performance of the algorithms, and Section 5 presents the results of these simulations. Section 6 offers a discussion and proposes different scenarios in which the new algorithms could be used alongside future avenues.

## 2. Amplitude-Particle Swarm Optimisation (A-PSO)

*Particle Swarm Optimisation Theory*

In PSO, each robot is capable of measuring the fitness of its current location and of remembering the previous location with the highest fitness [6]. At each timestep $k$, each robot $i$ communicates its own personal best location $\mathbf{y}^i[k]$ to the rest of the swarm, and the best among them is selected as the global best location $\mathbf{y_g}[k]$.

The spatial behaviour of each robot can be controlled using any PSO variant dedicated towards the control of physical robotic swarms [13–19]. For this paper, it is characterised by the following position and velocity update equations [15], which introduce maximum velocity limitation to the motion of the swarm,

$$\mathbf{x}^i[k+1] = \mathbf{x}^i[k] + \Delta t \, \mathbf{u}^i[k+1], \tag{1}$$

and

$$\mathbf{u}^i[k+1] = \omega \mathbf{u}^i[k] + c_1 \mathbf{r_1} \circ \mathrm{sgn}(\mathbf{y}^i[k] - \mathbf{x}^i[k]) + c_2 \mathbf{r_2} \circ \mathrm{sgn}(\mathbf{y_g}[k] - \mathbf{x}^i[k]). \tag{2}$$

The vectors $\mathbf{x}^i[k]$ and $\mathbf{u}^i[k]$ are the position and velocity of robot $i$ at timestep $k$. The operator $\circ$ is the Schur product. The parameter $\omega \in [0,1)$ is called the inertia weight and it is used to bound the velocity of the robot [23]. The parameters $c_1 > 0$ and $c_2 > 0$ are the cognitive and social coefficients and control the tendency to move towards the personal best location or the global best location [23]. Each element $j$ of the vectors $\mathbf{r_1}$ and $\mathbf{r_2}$ is drawn from the uniform distribution,

$$\mathbf{r_1}^j, \mathbf{r_2}^j \sim U(0,1) \qquad 1 \leq j \leq d,$$

where $d$ is the number of dimensions of the environment. The parameter $\Delta t$ is the PSO controller loop delay (i.e., the update rate is $1/\Delta t$). Delays caused by inter-robot communication and processing of sensor inputs will lead to larger values of $\Delta t$. The sgn function in (2) introduces a maximum velocity limit into the PSO controller [15], given by

$$V = \frac{\hat{c}\sqrt{d}}{1 - \omega}. \tag{3}$$

The parameter $\hat{c}$ is the behaviour coefficient and is given by

$$\hat{c} = c_1 + c_2. \tag{4}$$

In PSO robotic applications, the fitness of a location is typically calculated using the average intensity of the observed signal. Let $f_A$ be the fitness function that needs to be maximised, such that the fitness of location $\mathbf{x}$ at time $t_k = k \, \Delta t$ is given by

$$f_A(t_k, \mathbf{x}) = \frac{1}{T} \int_{-T}^{0} |s(t_k + t, \mathbf{x})|^2 \, \mathrm{d}t, \tag{5}$$

where $s(t_k + t, \mathbf{x})$ is the observed signal at position $\mathbf{x}$ and $T$ is its duration. This type of PSO will be referred to as amplitude-PSO (A-PSO), due to the use of signal intensity (relating to signal amplitude) to assign fitness to a location.

Typically, in PSO algorithms, each robot remembers its personal best location until it discovers a new location of higher fitness. This permanence precludes the use of dynamic fitness functions (i.e., moving sources, mutable environments, noise, etc.) and is referred to as the *Outdated Memory Problem* [24]. To address this, a forgetting property can be introduced that forces the robots to select new personal best locations over time [25,26]. Fusing a typical forgetting function [27] with (5), the fitness $\hat{f}^i[k]$ of the personal best location $\mathbf{y}^i[k]$ of robot $i$ at timestep $k$ is given by

$$\hat{f}^i[k] = \left\{ \begin{array}{ll} f_A(t_k, \mathbf{x}^i[k]), & \text{for } f_A(t_k, \mathbf{x}^i[k]) > \hat{f}^i[k-1] \\ \hat{f}^i[k-1] \times e^{-a}, & \text{otherwise} \end{array} \right\} \tag{6}$$

where $a$ is a positive scalar. This describes the case when a new personal best location is found (i.e., $f_A(t_k, \mathbf{x}^i[k]) > \hat{f}^i[k-1]$), and when the old personal best location is maintained. The overall personal best and global best location selection process is described by Algorithm A1, where $\mathbf{y_g}$ and $\hat{f}_g$ are the global best location and its associated fitness (assumed to be communicated to all robots at each time step).

It is possible to identify the following two major weaknesses of A-PSO.

1. As A-PSO relies on the intensity of the received signal, its performance is dependent on the Signal-to-Noise Ratio (SNR). This in turn will limit the maximum range of the algorithm and its ability to follow the gradient of the fitness function.
2. Each robot can only calculate the fitness of its current position and therefore, the exploration capabilities of the swarm depend on its span—this is referred to as the *Diversity Loss Problem* [24]. When the swarm is concentrated in a small area, far away from the source, its exploration and convergence capabilities will be limited.

These weaknesses can be improved by including wavefield correlation into the A-PSO algorithm, and it is this idea that will be explored in the rest of this paper.

## 3. Wavefield Correlation PSO

Instead of measuring only the signal intensity with a single sensor it is possible to correlate the signals from a pair of sensors. In the presence of noise, the correlation of two signals can offer ways to address the negative effects of noise, theoretically achieving faster and over-time more consistent convergence towards the source. Additionally, the information collected from two sensors can be combined to estimate the direction to the source.

Cross-correlation is the mathematical process used to describe the similarity between two signals. Let $s_1(t)$ and $s_2(t)$ represent two different signals (i.e., hydrophone signals), given by

$$
\begin{aligned}
s_1(t) &= s(t) + n_1(t), \\
s_2(t) &= s(t + \tau_{lag}) + n_2(t),
\end{aligned}
\tag{7}
$$

where $s(t)$ is the received signal, $n_1(t)$ and $n_2(t)$ represent uncorrelated noise and $\tau_{lag}$ is the time difference between the signal being received by each sensor, as illustrated by Figure 1a. The cross-correlation of the two signals is described by

$$
R(\tau) = \int_{-\infty}^{\infty} s_1(t) s_2(t + \tau) \, \mathrm{d}t,
\tag{8}
$$

where $\tau$ is the lag. In a robotic scenario, cross-correlation can be used to calculate the angle of arrival [28] of a signal at a pair of wavefield sensors (e.g., hydrophones). The value of $\tau$ that maximises $R$ can be associated to $\tau_{lag}$ using

$$
\tau_{lag} = \underset{\tau}{\mathrm{argmax}} \{ R(\tau) \},
\tag{9}
$$

thereby enabling identification of the delay between the time of arrivals of the two signals.

When the separation of the two sensors $D$ and the propagation speed of the received signal $c$ are known, $\tau_{lag}$ can be used to calculate the angle of arrival $\alpha$ of the signal and therefore the direction towards the source. Assuming that the sensors are placed at the front and back of the robot, the angle of arrival $\alpha$ is given by

$$
\alpha = \cos^{-1}\left( \frac{\tau_{lag} \, c}{D} \right).
\tag{10}
$$

The angle of arrival of the signal is calculated with respect to the orientation of the robot. There always exist two candidate directions of arrival (directional rays) as shown in Figure 1b given by

$$
\mathbf{r}_+[k] = \begin{bmatrix} \cos(\alpha + o[k]) \\ \sin(\alpha + o[k]) \end{bmatrix} \qquad \mathbf{r}_-[k] = \begin{bmatrix} \cos(-\alpha + o[k]) \\ \sin(-\alpha + o[k]) \end{bmatrix},
\tag{11}
$$

respectively, where $o[k]$ is the orientation of the robot at timestep $k$. In this way, when the robot is facing the source, $\mathbf{r}_+[k] \approx \mathbf{r}_-[k]$ and both directional rays point towards the same direction.

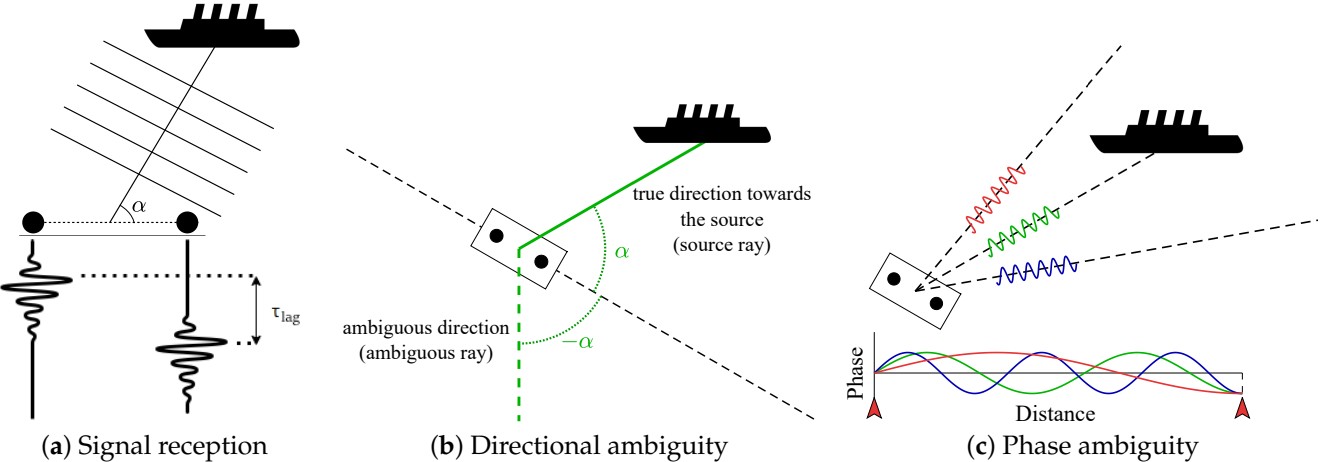

(**a**) Signal reception      (**b**) Directional ambiguity      (**c**) Phase ambiguity

**Figure 1.** Two sensors (black circles) on a single robot receive an incoming signal emitted by an external source (ship), at an angle $\alpha$. In (**a**), the signal is first received by the right-most sensor and then by the left-most one after a delay of $\tau_{lag}$, which can be used to estimate the angle $\alpha$ in (**b**). Phase ambiguities are related to the sensor separation, as shown in (**c**) where the phase lines show how all three signals can result in the same phase difference for the two sensors.

Additional ambiguities can be introduced due to the sensor separation $D$. This is because $\tau_{lag}$ depends on the phase difference of the signals from the two sensors. When the received signals are narrowband, it is possible that they will have the same phase difference when arriving at the robot from different angles, as shown in Figure 1c. To avoid this, the sensor separation is limited to $D \leq \lambda_c/2$, where $\lambda_c$ is the wavelength corresponding to the central frequency of the signal [28]. Furthermore, as the bandwidth of the signal increases, these types of ambiguities are generally reduced. This is because the signal consists of additional lower frequencies (higher wavelengths) that can be used in place of $\lambda_c$ to resolve ambiguities.

The non-normalised correlation coefficient $\rho$ is another metric provided by cross-correlation that shows the similarity between the two signals, given by

$$\rho = R(\tau_{lag}) = \max_\tau \{R(\tau)\}. \tag{12}$$

$\rho$ can be used to enhance the SNR of a received signal [29,30]. The value of $\rho$ depends on the intensity of $s(t)$ (as the intensity of $s(t)$ drops, e.g., due to attenuation, the value of $\rho$ drops as well).

### 3.1. Cross-Correlation Particle Swarm Optimisation (X-PSO)

Since the non-normalised correlation coefficient $\rho$ scales with the intensity of the received signal, it can be used in the same way that signal intensity is used in A-PSO. This can be achieved by replacing $f_A$ with $f_X$ given by,

$$f_X(t_k, \mathbf{x}) = \max_\tau \left\{ \int_{-T}^{0} s_1(t_k + t, \mathbf{x}) s_2(t_k + t + \tau, \mathbf{x}) \, \mathrm{d}t \right\}, \tag{13}$$

Therefore, the fitness $\hat{f}^i[k]$ of the personal best location $\mathbf{y}^i[k]$ of robot $i$ at timestep $k$ is given by (6) using $f_X(t_k, \mathbf{x}^i[k])$ instead of $f_A(t_k, \mathbf{x}^i[k])$. This approach is called cross-correlation PSO (X-PSO). The overall personal and global best selection process is described by Algorithm A2.

The advantage of X-PSO is that making use of the non-normalised correlation coefficient $\rho$ of two different signals can reduce the effects of noise, boosting SNR.

### 3.2. Bearing Particle Swarm Optimisation (B-PSO)

It is possible to use the directional rays to estimate the location of the source in an approach called bearing-PSO (B-PSO). In A-PSO and X-PSO, fitness is assigned to the current location of the robot. However, in B-PSO, fitness is assigned to locations elsewhere indicated by directional ray intersections.

In theory, at each timestep each robot could consider all intersections with all other robots of the swarm to maximise information gain, but this would require communication with all other robots, greatly increasing communication delays. Instead, at each timestep, each robot $i$ receives the directional rays of only one other random robot $j$ and combines it with its own to calculate ray intersections [31]. Combining each ray of robot $i$ with each ray of robot $j$, can result from zero to four intersections at any timestep. The aim of robot $i$ is to select the intersection that more closely approximates the location of the source.

Assuming that the SNR is significantly greater than 0 dB then on average at least one directional ray from each robot will point towards the source. Consider a single robot rotating about its central axis; one ray will remain pointing towards the source whilst the other (the ambiguous $-\alpha$ ray in Figure 1b) will rotate throughout 360 degrees. Extending to a swarm of robots, the ambiguous (rotating) rays will likely lead to ray intersections that are close to the robots (distributed around the swarm), whilst the stable source rays are likely to lead to intersections that are distant (i.e., distributed around the source). Therefore, to increase the probability that an intersection of two unambiguous rays (i.e., those that point towards the source) is identified, robot $i$ selects the *furthest* of its intersections from its current position.

A drawback of this process is that there is a risk that the selected intersections may be beyond the source or behind the swarm (especially for low SNR). Therefore, the most distal intersection (once selected) $\mathbf{p_s}^i$ is assigned a fitness $f_B$ based on its distance from robot $i$ given by

$$f_B(\mathbf{x}^i, \mathbf{p_s}^i) = \frac{1}{|\mathbf{x}^i - \mathbf{p_s}^i|}. \tag{14}$$

where $\mathbf{x}^i$ is the location of robot $i$. Thus, locations that are far away are progressively down weighted.

As with A-PSO and X-PSO, the selected intersection replaces the personal best location of robot $i$ if its fitness is higher than the previous personal best location. Note that maximising the fitness function $f_B$ does not necessarily imply convergence to the source. Thus, a forgetting function is used to ensure that the personal best location is updated as robot $i$ moves towards the source. In contrast to A-PSO and X-PSO, the forgetting function is fundamental for the correct operation of B-PSO. Therefore, the fitness $\hat{f}^i[k]$ of the personal best location $\mathbf{y}^i[k]$ of robot $i$ at timestep $k$ is given by (6) using $f_B(\mathbf{x}^i[k], \mathbf{p_s}^i[k])$ instead of $f_A(t_k, \mathbf{x}^i[k])$.

Finally, to locate the source the centroid of the personal best locations (i.e., the centroid of the selected and weighted intersections) may be computed and used as the global best location $\mathbf{y_g}$,

$$\mathbf{y_g}[k] = \frac{\sum_{i=1}^{M} \mathbf{y}^i[k]}{M}, \tag{15}$$

where $M$ is the total number of robots. The overall process is described by Algorithm A3.

B-PSO selects personal best locations that are distant from the swarm, thus fostering an exploration behaviour that addresses Weakness 2 (i.e., its exploration capabilities are not limited by the span of the swarm). However, B-PSO may still be susceptible to distal intersections caused by ambiguous rays (especially if they are located behind the swarm), since they may not be immediately replaceable due to low fitness. Therefore, a more direct

way of excluding such intersections using correlation measurements is considered in the next approach.

### 3.3. Cross-Correlation-Bearing Particle Swarm Optimisation (XB-PSO)

In order to identify intersections caused by ambiguous rays, X-PSO and B-PSO are now combined to form a cross-correlation-bearing PSO (XB-PSO) approach.

In XB-PSO both correlation coefficient $\rho$ and directional ray information of robot $j$ are known by robot $i$, and it is assumed that $\rho$ is large close to the source. Thus, each intersection $\mathbf{p}^{ij}$ formed by directional rays of robots $i$ and $j$ can be validated (as pointing towards the source) using either of the following conditions

$$|\mathbf{x}^i[k] - \mathbf{p}^{ij}[k]| < |\mathbf{x}^j[k] - \mathbf{p}^{ij}[k]| \quad \text{and} \quad \rho^i[k] > \rho^j[k] \tag{16a}$$

$$|\mathbf{x}^i[k] - \mathbf{p}^{ij}[k]| > |\mathbf{x}^j[k] - \mathbf{p}^{ij}[k]| \quad \text{and} \quad \rho^i[k] < \rho^j[k], \tag{16b}$$

where $\rho^i[k]$ and $\rho^j[k]$ are the non-normalised correlation coefficients of robots $i$ and $j$, respectively at timestep $k$. Intersections that do not satisfy either of these conditions are likely intersections of ambiguous rays and are therefore ignored.

From here, the process is identical to B-PSO, where the selected intersection $\mathbf{p_s}^i$ is assigned a fitness using (14). This fitness is compared to the fitness $\hat{f}^i$ of the personal best location and if it is higher, $\mathbf{p_s}^i$ becomes the new personal best location. Furthermore, the selection of the global best location is performed in the same manner as B-PSO, as described by (15). The overall personal and global best location selection process for XB-PSO is described by Algorithm A4.

## 4. Simulated Environment

In order to evaluate the efficacy of the proposed fitness assignment methods a series of simulations were performed using MATLAB. The simulations were framed in a marine defense application in which the goal is to detect and localise uncrewed underwater vehicles (UUV) [32–35]. The simulations consist of a single static UUV (the source of an acoustic signal) that is being localised by a swarm of uncrewed surface vehicles (USV) in a moderate sea state. Defining this scenario enables the selection of several key parameters given in Table 1, and are chosen primarily for the purposes of illustration by example. Whilst this scenario constrains the motion of the USVs to 2D there is nothing that precludes the use of higher dimensions.

The simulated environment is a 2D infinite plane representing the water surface, and there is assumed to be no friction and no body inertia. Acoustic signals are modelled with constant propagation at 1500 m/s, consistent with acoustic propagation close to the water surface, below any significant thermocline and in non-polar regions [36]. Multipath effects, such as between the water surface and the ocean floor are neglected.

### 4.1. Robots

The simulated robots (USVs) are equipped with two hydrophones located at the bow and stern. The swarm is initialised at random positions inside a circular area of radius $d_0$ around the origin, which represents deployment from a stationary ship. The velocity of each robot is directly controlled by the PSO controller defined in (2). The robots are not forced to maintain a minimum separation from each other and collisions are ignored.

### 4.2. Source

A single source (a static UUV) is positioned at a distance $d_s$ from the origin. Convergence to the source is achieved when the centre of mass of the swarm (CoM) reaches a distance $d_c$ from the source at time $t_c$. The source emits a continuous acoustic signal of constant power spectral density (PSD$_s$ in units of dB re 1 μPa$^2$/Hz at 1 m) over a bandwidth $B$. This relates to the noise generated by a UUV motor considering a narrow bandwidth $B$ around the peak frequency $f_c$ of the power spectral density of the motor signal. For elec-

trically propelled UUVs the typical power spectral density peaks at 120 dB re $1\,\mu\text{Pa}^2/\text{Hz}$, in the range $f_c$ from several hundred Hz to a few kHz [37–39]. The simulated signal is therefore modelled using a filtered Gaussian process with central frequency $f_c$, bandwidth $B$ and Q-factor given by $Q = f_c/B$. The source level intensity $I_s$ (with units of $\text{Pa}^2$ at 1 m) of the simulated signal is given by

$$I_s = 10^{\frac{\text{PSD}_s}{10}} \times 10^{-12} \times B, \tag{17}$$

representing the area under the constant power spectral density curve between $f_c - B/2$ and $f_c + B/2$. Equation (17) can be found by first converting dB to a ratio, normalising for $1\,\mu\text{Pa}^2$ and integrating over $B$, assuming constant power spectral density over this range. Only the Q-factor will be provided in the results to allow the generalisation of the results.

The source signal attenuates depending on the distance travelled, such that the intensity of the signal at distance $r$ is given by

$$I = \frac{I_s}{r^2}. \tag{18}$$

Filtered additive white Gaussian noise (AWGN) is added to the signals with constant power spectral density over bandwidth $B$ ($\text{PSD}_n$ in units of dB re $1\,\mu\text{Pa}^2/\text{Hz}$) and average intensity $N$, approximating a moderate sea state [40]. The intensity $N$ with units of $\text{Pa}^2$ is related to $\text{PSD}_n$ in the same way as $I_s$ is related to $\text{PSD}_s$ using (17). Therefore, the SNR at distance $r$ from the source is given by

$$\text{SNR} = 10\log_{10}\left(\frac{I_s}{N}\right) - 10\log_{10}(r^2) \tag{19}$$

The SNR of the signals at the origin (i.e., the location of the initial deployment of the swarm) $\text{SNR}_0$ is found by setting $r = d_s$ in (19). Varying the value of $d_s$ thus varies $\text{SNR}_0$ and provides a mechanism for readily assessing convergence with different initial SNR.

### 4.3. Normalised Units

Normalised units are used to enable the generalisation of the results to multiple scenarios. Normalised time is described in *timesteps*, such that each timestep has size $\Delta$t seconds as defined in Section 2. Normalised distance is relative to a reference distance $R$, defined as the distance in metres at which $I = N$ (i.e., at $r = R$ m, SNR = 0 dB). Therefore, $R$ is given by

$$R = \sqrt{\frac{I_s}{N}} \tag{20}$$

Varying $\Delta$t varies the controller loop delays (i.e., different communication rate between robots), while varying $R$ enables the consideration of different signal intensities and noise levels.

**Table 1.** Parameter values to approximate a marine source localisation scenario.

| Parameter | Value | Justification |
|---|---|---|
| Timestep ($\Delta$t) | 1 s | A typical controller timestep size for robotic applications. |
| Noise PSD ($\text{PSD}_n$) | 60 dB re $1\,\mu\text{Pa}^2/\text{Hz}$ | Equivalent to a moderate sea state [40]. |
| Source PSD ($\text{PSD}_s$) | 120 dB re $1\,\mu\text{Pa}^2/\text{Hz}$ at 1 m | Equivalent to a typical uncrewed underwater vehicle [37,38]. |

**Table 1.** *Cont.*

| Parameter | Value | Justification |
|---|---|---|
| Reference distance ($R$) | 1000 m | Calculated using (20). |
| Source centre frequency ($f_c$) | 1 kHz | The central frequency of a typical uncrewed underwater vehicle [37]. |
| Maximum velocity ($V$) | 2 m/s | Typical uncrewed surface vehicle maximum speed range is 1.5 m/s to 5 m/s [41]). |
| Signal propagation speed ($c$) | 1500 m/s | Speed of sound in water [36]. |
| No. Robots ($M$) | 10 | Common swarm size in marine robotics [4]. |
| Starting radius ($d_0$) and convergence radius ($d_c$) | 50 m | Sufficient to accommodate 10 robots. |
| Forgetting function scaling parameter ($a$) | 1 | Frequent updating of personal best locations. |

## 5. Results

Simulations were run to validate the proposed algorithms and to evaluate their performance (convergence capability) in a range of different scenarios. In all simulations the parameters given in Table 1 were used and each scenario was repeated for ($N_r = 100$) repeats (selected as the nearest multiple of 10 above which the statistical significance of the results did not change), to account for the effects of the random noise processes and the random vectors in (2).

This section will describe the convergence speed of the algorithms, their maximum range and their spatial behaviour.

### 5.1. Initial Signal-to-Noise Ratio

In order to characterise the convergence capabilities of the proposed algorithms the initial SNR ($\text{SNR}_0$) was varied from 10 to $-5$ dB. The ability of the swarms to achieve convergence more rapidly, and at lower $\text{SNR}_0$, implies that they are more robust to low SNR and have a larger effective range. The Q-factor of the received signal was fixed at $Q = 1.5$ (i.e., one-octave bandwidth), the sensor separation for each robot was $D = \lambda_c/2$ and the time-bandwidth product of the cross-correlated signals was TBP $= 100$.

Figure 2 shows the median and 90% percentile ranges of the distance from the source to the CoM of the swarm as function of time for $\text{SNR}_0 = 10, 4, 0, -3$ and $-5$ dB. These results show that A-PSO and X-PSO can only converge when $\text{SNR}_0 \geq 10$ dB, with $t_c = 4500$ and $t_c = 3400$, respectively. The performance of A-PSO and X-PSO are comparable, although the slightly faster convergence of X-PSO does indicate a resilience to low SNR that improves Weakness 1 of A-PSO.

B-PSO and XB-PSO both show convergence at all $\text{SNR}_0$ values, with similar convergence behaviours. B-PSO has an average $t_c =$ of 1200, 2700 and 4400 and XB-PSO an average $t_c = 1100$, 2600 and 4400 for $\text{SNR}_0 = 10$, 4 and 0 dB, respectively. This result demonstrates that these algorithms are more robust to low SNR compared to both A-PSO and X-PSO, and thus have superior maximum range.

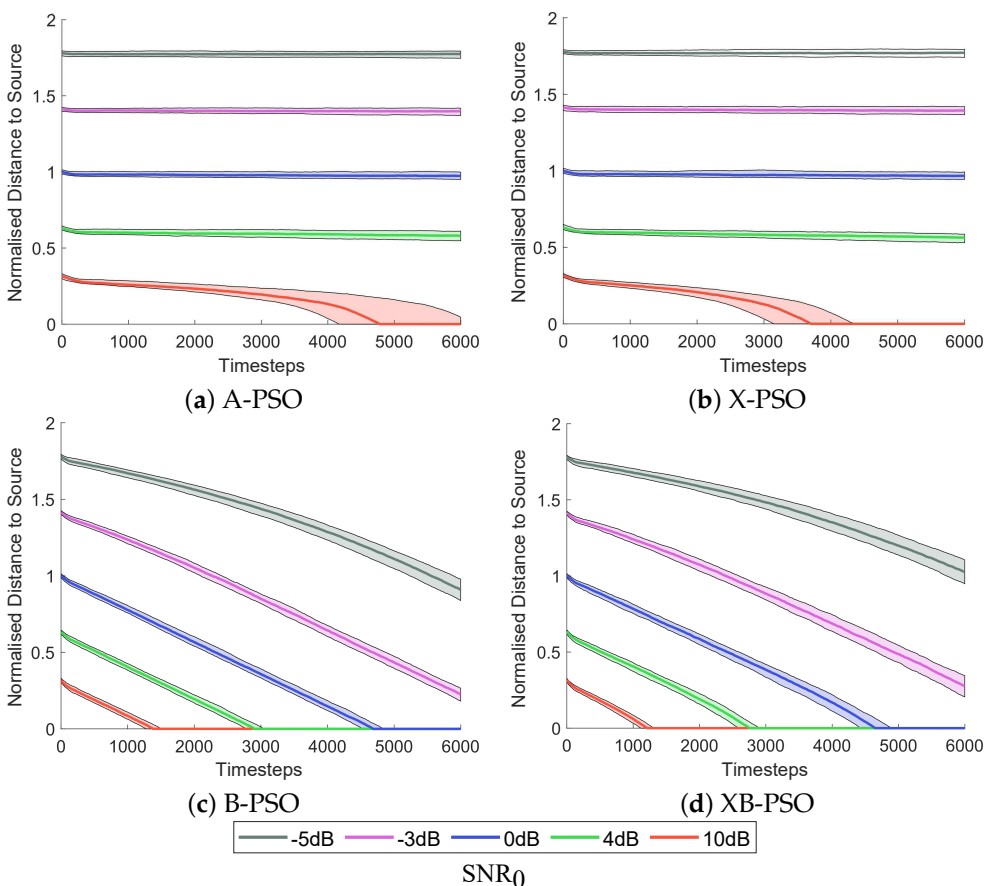

**Figure 2.** Distances of CoM to source at different initial SNR values ($SNR_0$). The solid lines represent the median distance from source over ($N_r = 100$ simulations) and the transparent areas the 90% percentile range.

Spatial Analysis

Since both B-PSO and XB-PSO employ personal best locations that distant from the swarm, it is expected that they will exhibit unique spatial motions compared to A-PSO and X-PSO that may explain their enhanced performance. In order to understand this behaviour the spatial routes followed by the swarms were analysed for an $SNR_0 = 10\,dB$, as at this $SNR_0$ all algorithms achieved convergence. A graphical representation of the spatial routes is shown in Figure 3, where the contours show the probability that the CoM of a swarm will pass through that location. The routes of three randomly selected swarms are also shown by the red, green and blue solid lines, while the corresponding dashed lines represent the projections of these routes on the vertical plane (i.e., Timesteps vs. Normalised distance X).

The results show that the different ways of selecting personal best locations (i.e., at the current location of a robot versus at a point defined by the intersection of directional rays) lead to very different spatial behaviour. A-PSO and X-PSO follow direct but zig-zagging routes, whereas B-PSO and XB-PSO move in an almost straight line towards the source.

The contours on the spatial axes of Figure 3 show similar results. The robots controlled by A-PSO and X-PSO are spread out until they get close to the source (occupancy probabilities (>50%) only appear at 0.25 normalised distance). On the other hand, B-PSO and XB-PSO exhibit less spreading and high occupancy probabilities appear at 0.1 normalised distance. These results provide some insight into the convergence performance seen in Figure 2. At 10 dB $SNR_0$, B-PSO and XB-PSO are more resilient to noise than A-PSO and X-PSO and so address Weakness 2 of A-PSO (i.e., they exhibit superior exploration capabilities that are not limited by the span of the swarm).

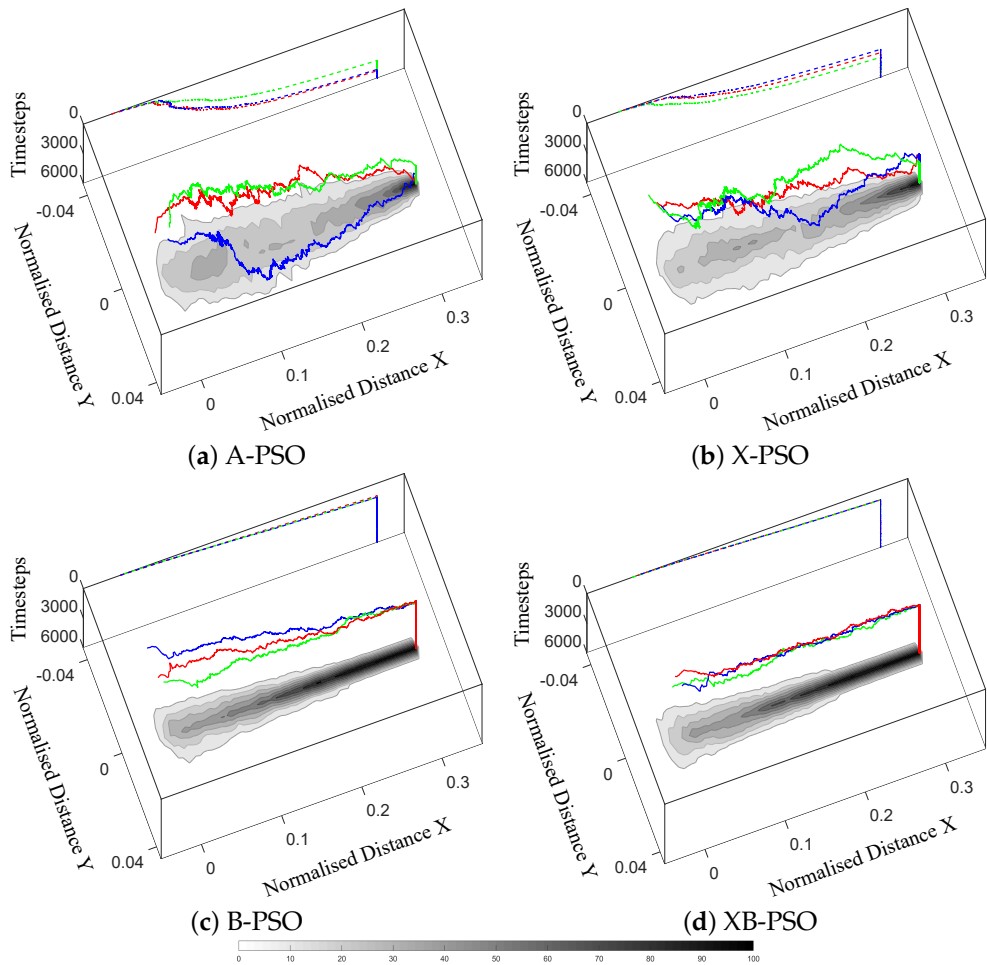

**Figure 3.** Graphical representation of the routes followed by the CoM of ($N_r$ = 100) swarms for each method. The image at the bottom plane of each graph shows contours of the number of swarms (CoM) that passed through that location. The red, green and blue lines represent the routes followed by three randomly selected swarms with respect to time. The corresponding dashed lines represent the 2D projections of these routes on the vertical plane (Timesteps vs. Normalised distance X).

*5.2. Q and Sensor Separation*

The results of Section 5.1 are presented in normalised time and distance to enable their generalisation to different scenarios. Adjusting $\Delta t$ allows the adaptation for different controller loop delays, while adjusting $R$ enables adaptation for different values of $PSD_n$ and $PSD_s$, (see (20)). Furthermore, the maximum velocity $V$ of the robots can be expressed in normalised distance per timestep and is inversely proportional to convergence time $t_c$ (i.e., $V \propto 1/t_c$). Therefore, as $V$ varies for different scenarios the convergence time $t_c$ will also vary inversely.

However, normalised time and distance cannot account for changes in Q (e.g., super-tankers have much lower $f_c$ than UUVs [42]) or the sensor separation D which can affect the convergence performance. Therefore, a sensitivity analysis was performed in which $Q$ and sensor separation $D$ were varied for $SNR_0 = 10\,dB$.

Figure 4 shows the normalised median time to convergence for different values of $Q$ and $\frac{D}{\lambda_c}$ computed over ($N_r$ = 100) repeats. Figure 4 shows that the performance of B-PSO and XB-PSO decreases as $Q$ increases and the signal approaches a sinusoid (narrowband), resulting in ambiguities in the lag output of the cross-correlation function, as discussed in Section 3, whereas A-PSO and X-PSO are insensitive. Despite the decrease in performance, XB-PSO still converges faster than B-PSO for all $Q$ values.

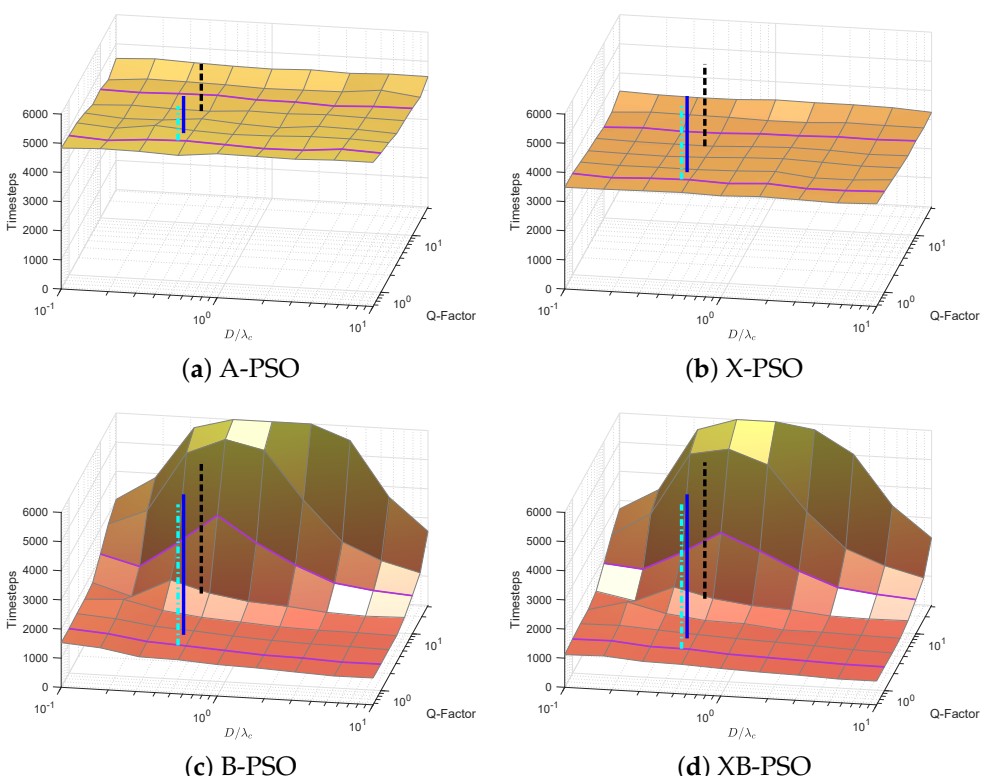

**Figure 4.** Normalised time needed for the CoM of a swarm to reach convergence (distance $d_c$ from the source). Each point represents the median performance over ($N_r = 100$) swarms. The purple horizontal lines represent the locations where Q is 1 and 10. The vertical blue solid line represents the simulations of Figure 2. The vertical black dashed and cyan dashed-dotted lines represent the Q-factors of the sound generated by two electrically-propelled UUVs, REMUS-100 and Odyssey IIb respectively [37–39]. Note that although a $D/\lambda_c$ axis is included for A-PSO, it does not affect the position of the sensor, since only one is used for that algorithm, located in the middle of the robot.

A-PSO and X-PSO are also insensitive to $D/\lambda_c$, whereas the performance of B-PSO and XB-PSO degrades as $D/\lambda_c$ approaches 1 for large values of $Q$. This is because phase ambiguities are introduced as the sensor separation increases beyond $\lambda/2$. However, as the separation approaches $10\lambda$, the performance of B-PSO and XB-PSO improves again because, even though more ambiguities are introduced, they are more evenly distributed around the robot. Therefore, as the robots change orientation the ambiguities are averaged out by the correct predictions of others.

## 6. Discussion

This paper has introduced three novel methods for location fitness assignment in PSO for the control of robotic swarms that extend and adapt intensity-based fitness (A-PSO) for applications of source localisation in scenarios that include wavefields. The results presented in Section 5 demonstrate that the novel algorithms (X-PSO, B-PSO and XB-PSO) address or improve the weaknesses of A-PSO. B-PSO and XB-PSO select personal best locations that are distant from the swarm, which enables them to converge with initial SNR as low as $-5$ dB. The convergence speed of the algorithms is affected by narrowband signals and sensor separations close to the wavelength of the received signal. XB-PSO converges faster than B-PSO, likely due to directional ray intersections that occur in the opposite direction from the source which cannot be effectively screened without also considering the signal intensity. However, the difference between the performance of the two methods is small.

For most applications, B-PSO and XB-PSO both offer a good choice of fitness assignment methods, especially when the sensor separation can be selected or broadband source signals are available. On the other hand, for specific applications where narrowband signals are used and the sensor separation is limited, X-PSO provides a more robust alternative. As examples, the results include the use cases of the REMUS-100 and Odyssey IIb, two common electrically-propelled UUVs. It can be seen that the fast convergence and low SNR requirements of B-PSO and XB-PSO make them ideal choices for the detection of relatively silent vehicles that emit low Q-factor signals.

The results make several key assumptions: that the emitted signal is continuous and of constant power spectral density over all frequencies, the robots operate in 2D environment with no friction and that the robots have unbounded rotational speed. Further studies are required to fully understand the implications of these assumptions.

Possible applications of the methods include any form of wavefield source localisation problem, including acoustic source localisation and radar localisation. Additionally, B-PSO and XB-PSO produce global best locations that are approximations of the source location, so can be used for source interaction (e.g., target entrapment and tracking). This capability is a major advancement to A-PSO that will enable its use in a wider range of source localisation scenarios.

## 7. Conclusions

This paper has presented three novel methods for location fitness assignment in PSO for the control of robotic swarms that consider the higher order statistics available in wavefield measurements. In simulations, they have been shown to outperform the original intensity-based PSO in terms of robustness to low signal-to-noise ratio (SNR) and convergence speed. The best performing method, cross-correlation bearing PSO (XB-PSO), is capable of converging to the source from as low as $-5$ dB initial SNR. The standard intensity-based PSO only manages to converge at 10 dB and at this SNR, XB-PSO converges 4 times faster

Additionally, B-PSO and XB-PSO may allow distant interaction with the source, a capability that is not offered by A-PSO or other standard fitness assignment methods. The algorithms have been demonstrated in simulation with an example scenario of a uncrewed underwater vehicle localisation. These novel algorithms significantly improve on the performance of A-PSO, and they may serve as inspiration for a new family of swarm robotic control algorithms that sense coherent wavefields.

**Author Contributions:** Conceptualization, G.R., A.H. and B.M.; methodology, G.R., A.H. and B.M; software, G.R.; validation, G.R.; formal analysis, G.R. and A.H.; investigation, G.R.; resources, B.M.; writing—original draft preparation, G.R.; writing—review and editing, B.M. and A.H.; visualization, G.R.; supervision, B.M. and A.H.; project administration, B.M.; funding acquisition, B.M. All authors have read and agreed to the published version of the manuscript.

**Funding:** This research was funded by both the UK Natural Environment Research Council (NERC) and Engineering and Physical Sciences Research Council (EPSRC) grant number NE/N012070/1. This research was partially funded by CMMI Cyprus Marine and Maritime Institute. CMMI was established by the CMMI/MaRITeC-X project as a "Center of Excellence in Marine and Maritime Research, Innovation and Technology Development" and has received funding from the European Union's Horizon 2020 research and innovation program under grant agreement No. 857586 and matching funding from the Government of the Republic of Cyprus.

**Institutional Review Board Statement:** Not applicable.

**Informed Consent Statement:** Not applicable.

**Data Availability Statement:** Not applicable.

**Conflicts of Interest:** The authors declare no conflict of interest. The funders had no role in the design of the study; in the collection, analyses, or interpretation of data; in the writing of the manuscript, or in the decision to publish the results.

## Appendix A. Algorithms

---

**Algorithm A1:** Personal and global best location selection for A-PSO.

---

1 **foreach** *robot* **do**
2 $\quad$ robot.$\mathbf{x} \leftarrow$ robot.GetCurrentPosition();
3 $\quad$ $s[] \leftarrow$ robot.ReadSignal(); // Get sensor reading
4 $\quad$ $f_A \leftarrow$ mean(abs$(s[])^2$); // Assign fitness
5 $\quad$ robot.UpdatePersonalBestLocation($f_A$);
6 **end**
7 // Calculate global best location
8 $\hat{f}_g \leftarrow 0$; // Reset global best location fitness
9 **foreach** *robot* **do**
10 $\quad$ **if** *robot.$\hat{f} > \hat{f}_g$* **then** // Update global best location
11 $\quad\quad$ $\hat{f}_g \leftarrow$ robot.$\hat{f}$;
12 $\quad\quad$ $\mathbf{y_g} \leftarrow$ robot.$\mathbf{y}$;
13 $\quad$ **end**
14 **end**

15 **Function** *UpdatePersonalBestLocation(self, fitness)* **:**
16 $\quad$ **if** *fitness > self.$\hat{f}$* **then** // Update personal best location
17 $\quad\quad$ self.$\hat{f} \leftarrow$ fitness;
18 $\quad\quad$ self.$\mathbf{y} \leftarrow$ self.$\mathbf{x}$;
19 $\quad$ **else** // Apply forgetting function
20 $\quad\quad$ self.$\hat{f} =$ self.$\hat{f} \times e^{-a}$;
21 $\quad$ **end**
22 **end**

---

---

**Algorithm A2:** Personal and global best location selection for X-PSO.

---

1 **foreach** *robot* **do**
2 $\quad$ robot.CalculateCorrelation();
3 $\quad$ $f_X \leftarrow$ robot.$\rho$; // Assign fitness to current location
4 $\quad$ robot.UpdatePersonalBestLocation($f_X$); // As in Algorithm A1
5 **end**
6 // Calculate global best location
7 $\hat{f}_g \leftarrow 0$; // Reset global best location fitness
8 **foreach** *robot* **do**
9 $\quad$ **if** *robot.$\hat{f} > \hat{f}_g$* **then** // Update global best location
10 $\quad\quad$ $\hat{f}_g \leftarrow$ robot.$\hat{f}$;
11 $\quad\quad$ $\mathbf{y_g} \leftarrow$ robot.$\mathbf{y}$;
12 $\quad$ **end**
13 **end**

14 **Function** *CalculateCorrelation(self)* **:**
15 $\quad$ self.$\mathbf{x} \leftarrow$ self.GetCurrentPosition();
16 $\quad$ $s_1[], s_2[] \leftarrow$ self.ReadSignals(); // Get sensor readings
17 $\quad$ $R[] \leftarrow$ xcorr($s_1[], s_2[]$); // Cross-correlate signals
18 $\quad$ $I_{lag} \leftarrow$ argmax($R[]$);
19 $\quad$ self.$\rho \leftarrow R[I_{lag}]$; // Calculate correlation coefficient
20 **end**

---

---

**Algorithm A3:** Personal and global best location selection for B-PSO.

---

**1** **foreach** *robot* **do**
**2** $\quad$ robot.CalculateCorrelation();
**3** **end**
**4** **foreach** *robot* **do** // Select personal best locations
**5** $\quad$ other $\leftarrow$ robot.SelectRandomOtherRobot(); // Uniformly random selection
**6** $\quad$ $\mathbf{p}[] = []$;// Initialise empty arrays of intersections
**7** $\quad$ // There can be up to 4 intersections
**8** $\quad$ **foreach** *ray* $\in$ *robot*.**r** **do**
**9** $\quad\quad$ **foreach** *otherRay* $\in$ *other*.**r** **do**
**10** $\quad\quad\quad$ $\mathbf{p}[]$.append(CalculateRayIntersections(robot.**x**, ray, other.**x**, otherRay));
**11** $\quad\quad$ **end**
**12** $\quad$ **end**
**13** $\quad$ $d[] = |$robot.$\mathbf{x} - \mathbf{p}[]|$; // Distances from robot to intersections
**14** $\quad$ $\mathbf{p_s} \leftarrow \mathbf{p}[\text{argmax}(d[])]$; // Select furthest intersection
**15** $\quad$ $f_B \leftarrow 1/\max(d[])$; // Assign fitness to selected intersection
**16** $\quad$ robot.UpdatePersonalBestLocation($f_B$, $\mathbf{p_s}$);
**17** **end**
**18** // Calculate global best location
**19** $\mathbf{y_g} \leftarrow [0,0]$; // Reset global best location
**20** **foreach** *robot* **do** // Sum personal best locations
**21** $\quad$ $\mathbf{y_g} \leftarrow \mathbf{y_g}+$robot.**y**;
**22** **end**
**23** $\mathbf{y_g} \leftarrow \mathbf{y_g}/M$; // Calculate centroid

**24** **Function** *CalculateCorrelation(self)* **:**
**25** $\quad$ self.**x** $\leftarrow$ self.GetCurrentPosition();
**26** $\quad$ $o \leftarrow$ self.GetCurrentOrientation();
**27** $\quad$ $s_1[], s_2[] \leftarrow$ self.ReadSignals(); // Get sensor readings
**28** $\quad$ $[R[], \tau[]] \leftarrow$ xcorr($s_1[],s_2[]$); // Cross-correlate signals
**29** $\quad$ $I_{lag} \leftarrow \text{argmax}(R[])$;
**30** $\quad$ self.$\rho \leftarrow R[I_{lag}]$; // Calculate correlation coefficient
**31** $\quad$ $\tau_{lag} \leftarrow \tau[I_{lag}]$; // Calculate lag
**32** $\quad$ $\alpha \leftarrow \text{acos}(\tau_{lag} * c/D)$;
**33** $\quad$ self.**r**[1] $\leftarrow [\cos(\alpha + o); \sin(\alpha + o)]$; // Calculate directional rays
**34** $\quad$ self.**r**[2] $\leftarrow [\cos(-\alpha + o); \sin(-\alpha + o)]$;
**35** **end**

**36** **Function** *UpdatePersonalBestLocation(self, fitness, $\mathbf{p_s}$)* **:**
**37** $\quad$ **if** *fitness* $>$ *self*.$\hat{f}$ **then** // Update personal best location
**38** $\quad\quad$ self.$\hat{f} \leftarrow$ fitness;
**39** $\quad\quad$ self.**y** $\leftarrow \mathbf{p_s}$;
**40** $\quad$ **else** // Apply forgetting function
**41** $\quad\quad$ self.$\hat{f} =$ self.$\hat{f} \times e^{-a}$;
**42** $\quad$ **end**
**43** **end**

---

---

**Algorithm A4:** Personal and global best location selection for XB-PSO.

```
1  foreach robot do
2  │  robot.CalculateCorrelation(); // As in Algorithm A3
3  end
4  foreach robot do // Select personal best locations
5  │  other ← robot.SelectRandomOtherRobot(); // Uniformly random selection
6  │  p[] = []; // Initialise empty arrays of intersections
7  │  // There can be up to 4 intersections
8  │  foreach ray ∈ robot.r do
9  │  │  foreach otherRay ∈ other.r do
10 │  │  │  p[].append(CalculateRayIntersections(robot.x, ray, other.x, otherRay));
11 │  │  end
12 │  end
13 │  d[] = |robot.x − p[]|; // Distances from robot to intersections
14 │  other_d[] = |other.x − p[]|; // Dist from other robot to intersections
15 │  for q ← length(p[]) to 0 do // Remove non-valid intersections
16 │  │  if robot.ρ > other.ρ and d[q] > other_d[q] then p[].remove(q);
17 │  │  else if robot.ρ < other.ρ and d[q] < other_d[q] then p[].remove(q);
18 │  end
19 │  d[] = |robot.x − p[]|; // Distances from robot to intersections
20 │  pₛ ← p[argmax(d[])]; // Select furthest intersection
21 │  f_B ← 1/max(d[]); // Assign fitness to selected intersection
22 │  robot.UpdatePersonalBestLocation(f_B, pₛ); // As in Algorithm 3
23 end
24 // Calculate global best location
25 y_g ← [0,0]; // Reset global best location
26 foreach robot do // Sum personal best locations
27 │  y_g ← y_g+robot.y;
28 end
29 y_g ← y_g/M; // Calculate centroid
```

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
