# Peer review of "Source Localisation Using Wavefield Correlation-Enhanced Particle Swarm Optimisation"

_robotics, doi:10.3390/robotics11020052_

Round 1
Reviewer 1 Report
a) Methods for solving dynamic optimization problems have been developed for a considerable time. For example, modifications of the simplex method for solving such problems are well known. I believe that the introduction should include a review of known methods for solving dynamic optimization problems.
b) The significance of the work would be greatly strengthened by comparing the efficiency of the algorithms proposed by the authors with some known algorithms for solving such problems.

Author Response
We thank the reviewer for their suggestion of alternative dynamic optimisation solvers. We chose to use PSO because of its inherent swarm intelligence characteristics that enable it to be used for the control of robotic swarms. Although we employ techniques for dynamic optimisation to deal with signal noise, our focus is the convergence of the robotic swarm towards the source. Therefore, we believe that mentioning alternative algorithms that do not employ swarm intelligence would not be suitable for our scope.
Regarding the comparison with current literature, we are comparing our proposed fitness assignment approaches with the standard fitness assignment approach used in the control of robotic swarms. Although we use one PSO variant to implement these approaches, any other PSO variant could have been used as well. We understand that this was not clear previously and therefore we have made changes to make it clearer.
In lines 41-57: We added several PSO variants that are used for the control of robotic swarms. We mention at the end that a different approach to assigning fitness could be used in all of them. Therefore, our proposed methods of assigning fitness could be combined with any of the mentioned PSO variants.
In line 88: We added that despite using a specific PSO variant for the motion control of the swarm, any other PSO variant could have been used instead.
Reviewer 2 Report
The manuscript presents the research addressing the source localization problem based on the PSO algorithm, with three novel methods: X-PSO, B-PSO, and XB-PSO.
Specifically, the X-PSO method proposes another fitness function (Eq. 13) to evaluate the robot position instead of using Eq. 5 as in a published article; In the B-PSO method, Eq.14 is used; The XB-PSO method is a combination of the two methods above. Simulations were presented to confirm the effectiveness of the proposed methods.
Obviously, the manuscript focuses on proposing and improving the fitness function - a method of evaluating the current robot position, and the mechanism of the PSO used is unchanged.
From the swarm and evolutionary computation perspective, it's not really three novel versions of PSO but instead new proposed fitness functions - where they're seen as black boxes when developing novel algorithm variants.
Therefore, sentences like: "...in this paper three novel PSO algorithms are introduced…" in the abstract will confuse readers.
Besides, the manuscript is partially missing the pseudocode.
Authors are encouraged to refine the manuscript slightly to avoid that issue.
Author Response
We thank the reviewer for pointing out how the manuscript was confusing, and we agree with their comments. Indeed, we are not proposing new PSO variants, but we propose new approaches for location fitness assignment. We have made changes to make this clearer and we added a variety of actual PSO variants in the introduction for completeness. We also made sure to clarify that our proposed fitness assignment approaches can be used by any of the presented PSO variants. We also fixed the problem with the missing pseudocodes.
In lines 41-57: We added several PSO variants that are used for the control of robotic swarms. We mention at the end that a different approach to assigning fitness could be used in all of them. Therefore, our proposed methods of assigning fitness could be combined with any of the mentioned PSO variants.
In line 88: We added that despite using a specific PSO variant for the motion control of the swarm, any other PSO variant could have been used instead.
We have also replaced every mention of “novel PSO algorithms” with “novel approaches for location fitness assignment in PSO”
Reviewer 3 Report
The paper presents novel algorithms for the control of robotic swarms. The main drawback is the well-known weakness of the naive PSO algorithm. As any algorithm can easily outperform naive PSO,
authors should consider adding a comparison with more recent and advanced optimization algorithms.
Author Response
We chose to use PSO because of its inherent swarm intelligence characteristics that enable it to be used for the control of robotic swarms. Although there exist advanced optimization algorithms that can surpass PSO in parameter optimization tasks, they are not always suitable for the control of robotic swarms in the way that PSO is used (i.e. for the direct control of the motion of the robots). Therefore, the use of such algorithms for performance comparison would lie outside the scope of this paper. Nevertheless, we added a variety of PSO variants that have been used for the control of robotic swarms in the introduction. We also made sure to clarify that our proposed methodology can be combined with any of the mentioned PSO variants, to show how our work complements existing literature.
In lines 41-57: We added several PSO variants that are used for the control of robotic swarms. We mention at the end that a different approach to assigning fitness could be used in all of them. Therefore, our proposed methods of assigning fitness could be combined with any of the mentioned PSO variants.
In line 88: We added that despite using a specific PSO variant for the motion control of the swarm, any other PSO variant could have been used instead.
Reviewer 4 Report
Dear authors,
The paper three novel PSO algorithms were described for the control of swarm robotics that consider the higher order statistics available in wavefield measurements. The article is well written and very detailed. The experiments help to support the conclusions. However, some items in the article can be improved. First, it is considered that the background to deal with related literature and the state of the art can be improved by considering some works in the literature, as in the following articles:
[1] Yang, J., Wang, X., & Bauer, P. (2019). Extended PSO based collaborative searching for robotic swarms with practical constraints. IEEE Access, 7, 76328-76341.
[2] Bakhale, M., Hemalatha, V., Dhanalakshmi, S., Kumar, R., & Siddharth Jain, M. (2020). A dynamic inertial weight strategy in micro PSO for swarm robots. Wireless Personal Communications, 110(2), 573-592.
[3] Lopes, H. J., & Lima, D. A. (2020, October). Cellular Automata in path planning navigation control applied in surveillance task using the e-Puck architecture. In 2020 IEEE International Conference on Systems, Man, and Cybernetics (SMC) (pp. 1117-1122). IEEE.
[4] Lima, D. A., Tinoco, C. R., & Oliveira, G. (2016, September). A cellular automata model with repulsive pheromone for swarm robotics in surveillance. In International Conference on Cellular Automata (pp. 312-322). Springer, Cham.
[5] Du, Y. (2020). A novel approach for swarm robotic target searches based on the DPSO algorithm. IEEE Access, 8, 226484-226505.
[6] Lima, D. A., & Oliveira, G. M. (2017). A cellular automata ant memory model of foraging in a swarm of robots. Applied Mathematical Modelling, 47, 551-572.
[7] Melo, R., Junior, C. S., & Victor, C. (2018). Performance analysis of a PSO-based algorithm for swarm robotics. In Proc. 9th Int. Multi-Conf. Complex., Inform. Cybern.(IMCIC) (pp. 1-7).
[8] Lopes, H. J., & Lima, D. A. (2021). Evolutionary Tabu Inverted Ant Cellular Automata with Elitist Inertia for swarm robotics as surrogate method in surveillance task using e-Puck architecture. Robotics and Autonomous Systems, 144, 103840.
[9] Xu, L., Song, B., & Cao, M. (2021). A new approach to optimal smooth path planning of mobile robots with continuous-curvature constraint. Systems Science & Control Engineering, 9(1), 138-149.
[10] Sancaktar, I., Tuna, B., & Ulutas, M. (2018). Inverse kinematics application on medical robot using adapted PSO method. Engineering science and technology, an international journal, 21(5), 1006-1010.
[11] Lima, D. A. et al. Coordination, synchronization and localization investigations in a parallel intelligent robot cellular automata model that performs foraging task. In ICAART (2017).
It would be interesting for the authors to consider newer work for optimization and localization using various bio-inspired techniques.
Furthermore, it would be interesting to establish a parallel between the proposed work and the works in the literature, focusing on the main differences between the proposals. This greatly improves the work from a scientific point of view.
Author Response
We thank the reviewer for the list of articles provided. Although we find the pheromone-based methods very interesting and suitable for area searching and path planning, we chose not to mention them in the manuscript since their direct comparison with PSO-based methods is not trivial due to the different way that they operate. Nevertheless, we have added in the introduction a variety of PSO variants that have been proposed for the control of robotic swarms, including some of the provided PSO-based articles.
We also made sure to clarify how our work is related and can be combined with these methods. Since we focus on different ways that location fitness assignment can be performed, our method can be easily combined with existing PSO variants that focus more on the motion dynamics of the robots and the amount of information exchanged.
In lines 41-57: We added several PSO variants that are used for the control of robotic swarms. We mention at the end that a different approach to assigning fitness could be used in all of them. Therefore, our proposed methods of assigning fitness could be combined with any of the mentioned PSO variants.
In line 88: We added that despite using a specific PSO variant for the motion control of the swarm, any other PSO variant could have been used instead.